# Chronic bronchitis without airflow obstruction, asthma and rhinitis are differently associated with cardiovascular risk factors and diseases

Marcello Ferrari[1], Elia Piccinno[1], Alessandro Marcon[2]*, Pierpaolo Marchetti[2], Lucia Cazzoletti[2], Pietro Pirina[3], Salvatore Battaglia[4], Amelia Grosso[5], Giulia Squillacioti[6], Leonardo Antonicelli[7], Giuseppe Verlato[2], Giancarlo Pesce[8]

1 Department of Medicine, Unit of Respiratory Medicine, University of Verona, Verona, Italy, 2 Unit of Epidemiology and Medical Statistics, Department of Diagnostics and Public Health, University of Verona, Verona, Italy, 3 Unità Operativa di Pneumologia, Dipartimento di Scienze Mediche, Chirurgiche e Sperimentali, Università degli Studi di Sassari, Sassari, Italy, 4 Dipartimento Universitario di Promozione della Salute, Materno Infantile, Medicina Interna e Specialistica di Eccellenza "G. D'Alessandro"(PROMISE), Università di Palermo, Palermo, Italy, 5 Division of Respiratory Diseases, IRCCS "San Matteo" Hospital Foundation, University of Pavia, Pavia, Italy, 6 Department of Public Health and Pediatrics, University of Turin, Turin, Italy, 7 Allergy Unit, Department of Internal Medicine, Azienda Ospedaliero-Universitaria Ospedali Riuniti, Ancona, Italy, 8 Epidemiology of Allergic and Respiratory Diseases Department (EPAR), Institut Pierre Louis d'Épidémiologie et de Santé Publique (IPLESP), Sorbonne Université, INSERM UMR-S 1136, Paris, France

* alessandro.marcon@univr.it

**Data Availability Statement:** All relevant data are within the manuscript and its Supporting Information files.

## Abstract

### Background and objectives

Cardiovascular and respiratory diseases can frequently coexist. Understanding their link may improve disease management. We aimed at assessing the associations of chronic bronchitis (CB), asthma and rhinitis with cardiovascular diseases and risk factors in the general population.

### Methods

We used data collected in the Gene Environment Interactions in Respiratory Diseases study, an Italian multicentre, multicase-control study. Among 2463 participants (age 21–86, female 50%) who underwent standardized interviews, skin prick and lung function tests, we identified 254 cases of CB without airflow obstruction, 418 cases of asthma without CB, 959 cases of rhinitis alone, and 832 controls. The associations of respiratory diseases with reported cardiovascular risk factors (lifestyles, hypertension, dyslipidaemia), heart disorders (myocardial infarction, coronary thrombosis, angina, aorta or heart surgery) and intermittent claudication were estimated through relative risk ratios (RRR) by multinomial logistic regression models.

**Funding:** The GEIRD study was funded by the Cariverona foundation, the Italian Ministry of Health, Chiesi Farmaceutici SpA, and the Agenzia Italiana del Farmaco (AIFA). The funders had no role in study design, data collection and analysis, decision to publish, or preparation of the manuscript.

**Competing interests:** The authors have declared that no competing interests exist.

**Abbreviations:** BMI, body mass index; CB, chronic bronchitis; COPD, chronic obstructive pulmonary disease; $FEV_1$, forced expiratory volume in 1 second; FVC, forced vital capacity; GEIRD, Gene Environment Interactions in Respiratory Diseases; LLN, lower limit of normal; $PD_{20}$, provocative dose causing a 20% decrease in $FEV_1$; RRR, relative risk ratios.

## Results

Compared to controls, CB cases were more likely to be heavy smokers, alcohol consumers, physically inactive, and to suffer from hypertension or dyslipidaemia; rhinitis cases were less obese but more likely to have hypertension. Asthma was significantly associated with current smoking. After adjusting for cardiovascular risk factors, heart disorders were associated with CB (RRR[95%CI]: 1.58[1.12–2.22]) and rhinitis (1.35[0.98–1.85]) and intermittent claudication was associated with CB (3.43[2.52–4.67]), asthma (1.51[1.04–2.21]) and rhinitis (2.03[1.34–3.07]).

## Conclusions

CB, asthma and rhinitis were associated with cardiovascular risk factors and diseases. In particular, CB shared with cardiovascular diseases almost all risk factors and was strongly associated with a higher risk of heart disorders and intermittent claudication.

## Introduction

Several previous studies have shown a significantly increased risk of cardiovascular diseases in COPD and clinicians have long recognized that cardiovascular diseases are the major contributor to morbidity and mortality in patients with COPD [1,2].

Other respiratory diseases have been associated with cardiovascular diseases. Large population studies showed that patients with chronic bronchitis (CB), a respiratory condition associated with a decline in lung function [3], have increased respiratory, cardiovascular and all-cause mortality [4–7].

Data on the association between asthma and cardiovascular diseases are conflicting [8–14], and few studies have addressed the relationship between rhinitis and cardiovascular diseases [15–17].

Previous research on the intriguing coexistence of respiratory and cardiovascular diseases generally focused on a single respiratory disorder, and few studies investigated the cardiovascular risk factors associated with airway illnesses. With this in mind, we aimed to investigate the association of cardiovascular diseases and cardiovascular risk factors with CB, asthma and rhinitis, by analysing data collected in the population-based Gene Environment Interactions in Respiratory Diseases (GEIRD) study.

## Methods

### Study design and selection of the cohort

GEIRD is a two-stage multicentre, multicase–control study carried out in Italy (www.geird.org) [18]. In the first stage, new random samples of adults (GEIRD 20–64 years) and elderly subjects (GEIRD 65–85 years), male/female = 1/1, or pre-existing randomly sampled cohorts from the general population (ISAYA, ECRHS Italy and ECRHS III) were surveyed for respiratory symptoms between 2006 and 2010 using a postal questionnaire, as reported in Table 1 and described elsewhere [19]. Overall, 14,513 subjects from the centres of Verona, Pavia, Torino, Ancona, Sassari answered the questionnaire (response rate: 59%). All the subjects who reported symptoms suggestive of asthma or chronic bronchitis, a random sample (30%) of the subjects who reported rhinitis or hay fever and a random sample (40%) of the subjects who did not report respiratory symptoms, diagnoses or hospitalizations were invited to clinics. Additionally, a sample of 439 subjects from Palermo was invited. Overall, 7,025 subjects were

**Table 1. Study population by centre and cohort.**

| Center | Cohort | Invited to stage 2 | Participating in stage 2 | Included in the analysis | Females (%) | Age, years (mean ± SD) |
|---|---|---|---|---|---|---|
| Verona | ECRHS III | 185 | 98 | 95 | 47.4 | 54.2 ± 7.6 |
| | GEIRD 20–64 | 2,961 | 1,329 | 1,165 | 52.2 | 44.5 ± 10.2 |
| | ISAYA | | | | | |
| | ECRHS Italy | | | | | |
| | GEIRD 65–84 | 591 | 132 | 97 | 34.0 | 71.8 ± 2.8 |
| Turin | ECRHS III | 178 | 76 | 69 | 55.1 | 53.9 ± 6.3 |
| | GEIRD 20–64 | 589 | 359 | 282 | 52.1 | 46.8 ± 10.3 |
| Pavia | ECRHS III | 186 | 95 | 86 | 53.5 | 57.2 ± 6.7 |
| | GEIRD 20–64 | 489 | 241 | 204 | 62.3 | 50.9 ± 11.0 |
| Ancona | GEIRD 20–44 | 575 | 99 | 91 | 53.8 | 42.3 ± 5.3 |
| Sassari | GEIRD 65–84 | 439 | 189 | 122 | 32.0 | 74.2 ± 4.3 |
| | ISAYA | 393 | 230 | 207 | 49.3 | 45.0 ± 6.9 |
| Palermo | GEIRD 65–84 | 439 | 63 | 45 | 33.3 | 75.1 ± 4.7 |
| **Overall** | | **7,025** | **2,911** | **2,463** | **50.7** | **49.5 ± 12.8** |

ECRHS, European Community Respiratory Health Survey; GEIRD, Gene Environment Interactions in Respiratory Diseases; ISAYA, Italian Study on Asthma in Young Adults.

invited to GEIRD stage-2 [20]. Between 2007 and 2015, the participants in GEIRD stage-2 were 2,911 subjects (participation rate: 41%). In all centres, clinical examinations were carried out following standardised protocols. Ethical approval was obtained from the following ethics committees: Verona, Comitato Etico per la Sperimentazione dell'Azienda Ospedaliera Istituti Ospitalieri di Verona; Turin, Comitato Etico dell'Azienda Sanitaria Locale TO/2 di Torino; Pavia, Comitato di Bioetica della Fondazione IRCCS Policlinico San Matteo di Pavia; Ancona, Comitato Etico dell'Azienda Ospedaliero-Universitaria Ospedali Riuniti di Ancona; Sassari, Comitato di Bioetica dell'Azienda Sanitaria Locale di Sassari; Palermo, Comitato Etico dell'Azienda Ospedaliera Ospedali Riuniti Villa Sofia Cervello. Written consent was obtained from each participant.

## Clinical measurements

Forced expiratory volume in 1 s ($FEV_1$) and forced vital capacity (FVC) were measured according to the American Thoracic Society reproducibility criteria [21]. Lung function values were expressed as a percentage of predicted values, and the lower limit of normal LLN for the $FEV_1$/FVC was calculated according to Quanjer [22]. Spirometry was performed again 10 min after the administration of 400 μg salbutamol in subjects with $FEV_1$/FVC <70% or <LLN. The subjects with $FEV_1$/FVC ≥70% and ≥LLN underwent the methacholine challenge, according to a standardized protocol [23]. A subject's test was positive if $FEV_1$ decreased by 20% at a maximum cumulative dose ≤1 mg methacholine ($PD_{20}$ ≤1).

The subjects were skin tested for a panel of 14 aeroallergens [20]. A subject was considered to be atopic if positive to one or more of the tested allergens.

## Identification of cases and controls in clinics

Based on the symptoms reported and the results of the clinical tests, 2,463 subjects were hierarchical classified into four groups: CB (n = 254), asthma (n = 418), rhinitis (n = 959), and controls (n = 832). The cases and control groups were defined as follows:

- **cases of chronic bronchitis** (CB): subjects with self-reported cough and phlegm for the most of days in 3 consecutive months, during 2 years, with post-bronchodilator $FEV_1/FVC \geq 70\%$ and $\geq LLN$.

- **cases of asthma**: subjects without CB who had 1) self-reported asthma, plus one among 1.1) having had an asthma attack in the last 12 months, 1.2) current use of medications for asthma; or 2) asthma-like symptoms or use of medicines for breathing problems in the last 12 months, plus one among 2.1) $PD_{20} \leq 1$ mg, 2.2) pre-bronchodilator $FEV_1/FVC < 70\%$ or $<LLN$ with a positive reversibility test (i.e. $FEV_1$ improvement $\geq 12\%$ and $\geq 200mL$ after 400μg of salbutamol);

- **cases of rhinitis**: subjects without CB and asthma who had one among 1) lifetime nasal allergies, including 'hay fever'; 2) lifetime problem with sneezing, or a runny or a blocked nose (without cold/flu); 3) recurrent nasal/eye symptoms in the presence of dust, pollens or animals.

- **controls**: subjects who were not cases and had both (i) pre-bronchodilator $FEV_1/FVC \geq 70\%$ and $\geq LLN$; and (ii) $FEV_1 > 80\%$ predicted.

Sixty-eight subjects with COPD, defined as having both persistent respiratory symptoms (dyspnoea, cough, and/or sputum production) and airflow limitation (post-bronchodilator $FEV_1/FVC < 70\%$ or $<LLN$), and 380 subjects who did not correspond to any of the definitions above or with missing values on key information were excluded from the analyses.

## Cardiovascular diseases

Two different self-reported doctor-diagnosed cardiovascular conditions were considered [24]:

1. **heart disorders**, defined as having any among coronary heart disease ('Did a physician tell you that you suffer from: myocardial infarction, coronary thrombosis, or angina?'), heart/ aortic surgery ('Have you ever undergone heart or aortic surgery?').

2. **intermittent claudication**. A subject was considered to have intermittent claudication if he/she answered yes to the question: 'Do you get a pain or discomfort in your legs when you walk?', plus he/she reported that it usually disappeared in 10 min or less when standing still [25]. Intermittent claudication was adopted as proxy of peripheral arterial disease [26].

## Covariates

Information on the following variables was collected during the clinical interview and was taken into account for the analyses: age, gender, school education as a proxy of the socio-economic status (low if had completed full-time education before the age of 16), smoking habits (lifetime non-smoker, ex-smoker, current smoker), daily alcohol intake (lifetime non-consumer, moderate ($\leq 15$ g/day), high ($>15$ g/day)), sedentary life (usually doing physical exercise less than once per month), and self-reported diabetes, hypertension or dyslipidaemia (high levels of cholesterol or triglycerides).

## Statistical analysis

The subjects' characteristics were summarized as percentages or means (SD). The Pearson's chi-squared test, Fisher's exact test and ANOVA were used to test differences across cases and controls ($\alpha = 0.05$). The associations of cardiovascular diseases/risk factors with respiratory diseases were estimated through relative risk ratios (RRR) obtained by multinomial logistic

regression models using the case/control indicator (0 = control, 1 = chronic bronchitis, 2 = asthma, 3 = rhinitis) as the dependent variable and cardiovascular diseases/risk factors as the main independent variable. Three models were fitted to the data: (i) adjusted for age and sex; (ii) further adjusted for smoking habits, alcohol consumption, body mass index (BMI, in categories <25, 25–30 and >30 kg/m$^2$), school education and physical activity; (iii) further adjusted for the comorbidity indicators (diabetes, hypertension, dyslipidaemia). Centre was considered a clustering factor and cluster-robust standard errors were used.

Statistical analyses were performed with STATA 13.1 (Stata Corp. College Station, TX, USA).

## Results

Overall, 2,463 subjects were classified as cases or controls and included in the study. The age ranged from 21 to 86 years, and 50.7% (n = 1,249) were females (Table 1).

As a result of our hierarchical definitions, cases of CB could also be affected by asthma (n = 115, 45.3%) as well as rhinitis (n = 205, 80.7%), and cases of asthma could also be affected by rhinitis (n = 354, 84.7%). Cases of asthma and CB had a significantly lower FEV$_1$ and FEV$_1$/FVC ratio than controls (Table 2). The proportion of subjects with atopy was significantly higher among cases of CB, asthma and rhinitis (57.8, 79.0 and 62.5%, respectively) compared to the control group (26.0%).

**Table 2. Distribution of risk factors, clinical characteristics and lung function measurements by cases-control status.**

| | Chronic bronchitis | Asthma | Rhinitis | Controls | p-value |
|---|---|---|---|---|---|
| | n = 254 | n = 418 | n = 959 | n = 832 | |
| **Sex** (females) | 145 (57.1) | 194 (46.4) | 496 (51.7) | 414 (49.8) | 0.048 |
| **Age** (year) | 48.4 ± 12.8 | 46.4 ± 12.2 | 49.7 ± 13.0 | 51.1 ± 12.6 | <0.001 |
| **Low education** | 65 (25.7) | 64 (15.6) | 152 (16.0) | 161 (19.5) | 0.001 |
| **Smoking habits** | | | | | <0.001 |
| never smokers | 112 (44.1) | 196 (46.9) | 490 (51.3) | 430 (54.7) | |
| ex-smokers | 58 (22.8) | 125 (29.9) | 295 (30.9) | 268 (32.3) | |
| current-smokers | 84 (33.1) | 97 (23.2) | 171 (17.9) | 133 (16.0) | |
| **Alcohol consumption** | | | | | 0.005 |
| no | 133 (53.4) | 231 (56.6) | 594 (63.0) | 543 (62.9) | |
| moderate (≤15 g/day) | 66 (26.5) | 115 (28.2) | 222 (23.5) | 210 (25.7) | |
| high (>15 g/day) | 50 (20.1) | 62 (15.2) | 127 (13.5) | 93 (11.4) | |
| **Sedentary life** | 163 (64.2) | 212 (50.8) | 488 (51.2) | 437 (52.5) | 0.002 |
| **BMI** | | | | | 0.388 |
| <25 kg/m$^2$ | 123 (50.6) | 224 (54.9) | 511 (55.8) | 317 (50.2) | |
| 25–30 kg/m$^2$ | 82 (33.7) | 127 (31.1) | 295 (32.2) | 292 (35.1) | |
| >30 kg/m$^2$ | 38 (14.7) | 57 (14.0) | 110 (12.0) | 122 (14.7) | |
| **Diabetes** | 9 (3.5) | 11 (2.6) | 35 (3.7) | 31 (3.7) | 0.766 |
| **Hypertension** | 80 (31.6) | 91 (21.8) | 243 (25.4) | 199 (24.0) | 0.032 |
| **Dyslipidaemia** | 97 (38.2) | 115 (27.7) | 277 (29.0) | 254 (30.5) | 0.022 |
| **FEV$_1$** (L) | 3.15 ± 0.87 | 3.24 ± 0.89 | 3.29 ± 0.83 | 3.32 ± 0.78 | <0.001 |
| **FVC** (L) | 4.02 ± 1.09 | 4.20 ± 1.15 | 4.05 ± 1.03 | 4.04 ± 0.96 | <0.001 |
| **FEV$_1$/FVC ratio (%)** | 78.6 ± 8.6 | 77.4 ± 6.8 | 81.3 ± 6.3 | 82.4 ± 6.0 | <0.001 |
| **FEV$_1$ (% of predicted)** | 96.2 ± 14.7 | 94.4 ± 14.0 | 100.9 ± 12.9 | 103.2 ± 12.3 | <0.001 |
| **Atopy** | 122 (57.8) | 278 (79.0) | 498 (62.5) | 177 (26.0) | <0.001 |

Data are presented as n (%) or mean ± standard deviation.

**Table 3. Relative risk ratios, with 95% confidence intervals, for the associations of demographics, cardiovascular risk factors and comorbidities with CB, asthma, and rhinitis.**

|  | Chronic bronchitis vs. Controls RRR (95% CI) | Asthma vs. Controls RRR (95% CI) | Rhinitis vs. Controls RRR (95% CI) |
|---|---|---|---|
| **Sex** (female vs. male) | **1.63 (1.31–2.04)** | 0.91 (0.76–1.10) | 1.09 (0.96–1.24) |
| **Age** (per year increase) | **0.96 (0.94–0.98)** | **0.96 (0.95–0.98)** | 0.99 (0.98–1.00) |
| **Education** (low vs. high) | **1.78 (1.24–2.54)** | 1.00 (0.66–1.50) | 0.82 (0.66–1.02) |
| **Smoking habits** |  |  |  |
| ex vs. never smokers | 0.78 (0.59–1.03) | 1.15 (0.91–1.46) | 1.01 (0.84–1.22) |
| current vs. never smokers | **2.00 (1.41–2.84)** | **1.52 (1.20–1.94)** | 1.16 (0.98–1.37) |
| **Alcohol consumption** |  |  |  |
| moderate vs. no | 1.27 (0.87–1.86) | 0.98 (0.74–1.28) | 0.90 (0.70–1.16) |
| high vs. no | **2.55 (1.61–4.03)** | 1.44 (0.99–2.11) | 1.22 (0.92–1.61) |
| **Sedentary vs. active life** | **1.59 (1.05–2.42)** | 1.02 (0.83–1.26) | 0.96 (0.73–1.26) |
| **BMI** |  |  |  |
| 25–30 vs. <25 kg/m$^2$ | 1.00 (0.78–1.30) | 0.91 (0.66–1.27) | **0.86 (0.79–0.94)** |
| >30 vs <25 kg/m$^2$ | 0.88 (0.64–1.21) | 0.99 (0.75–1.30) | 0.73 (0.52–1.03) |
| **Diabetes** (yes vs.no) | 0.91 (0.50–1.67) | 0.82 (0.38–1.79) | 1.12 (0.78–1.61) |
| **Hypertension** (yes vs.no) | **1.90 (1.14–3.15)** | **1.41 (1.06–1.89)** | **1.42 (1.13–1.77)** |
| **Dyslipidaemia** (yes vs.no) | **1.66 (1.19–2.31)** | 1.02 (0.84–1.24) | 0.95 (0.83–1.09) |

Adjusted for all the variables included in the table. RRR, relative risk ratio

The groups differed in the distribution of sex and age, as well as of most of lifestyle variables and cardiovascular risk factors: education, smoking habits, alcohol consumption, physical activity, hypertension, dyslipidaemia (Table 2). In particular, CB was significantly associated with female sex (RRR 1.63, 95%CI: 1.31–2.04), younger age (0.96, 0.94–0.98), and low education (1.78; 1.24–2.54) (Table 3).

Current smoking (2.00, 1.41–2.84), high alcohol consumption (2.55, 1.61–4.03) and sedentary life (1.59, 1.05–2.42) were also associated with an increased risk of CB. The risk of having asthma was higher in younger subjects (0.96, 0.95–0.98) and in current smokers (1.52, 1.20–1.94). The risk of having rhinitis was lower in overweight subjects (0.86, 0.79–0.94). Hypertension was significantly associated with CB (1.90, 1.14–3.15), asthma (1.41, 1.06–1.89) and rhinitis (1.42, 1.13–1.77). Dyslipidaemia was associated with CB (1.66, 1.19–2.31).

The crude prevalence of heart disorders was higher among cases of CB and rhinitis (14.2% and 12.1%) compared to controls and asthma cases (10.1% and 9.3%) (Fig 1).

After adjustment for all cardiovascular risk factors and comorbidities (Fig 2, model 3), heart disorders were significantly associated with CB (RRR, 95%CI: 1.58, 1.12–2.22; p = 0.009). A borderline association was also detected between heart disorders and rhinitis (RRR, 95%CI: 1.35, 0.98–1.85; p = 0.066).

The prevalence of intermittent claudication was higher in all the cases groups (10.6%, 4.8%, 6.0% in CB, asthma, and rhinitis groups, respectively) than in controls (3.3%) (Fig 1). In the fully adjusted model (Fig 2, model 3), intermittent claudication was significantly associated with a 3.5-fold higher risk of CB (RRR 3.43, 95%CI: 2.52–4.67; p<0.001), a 2-fold higher risk of rhinitis (2.03, 1.34–3.07; p<0.001), and a 1.5-fold higher risk of asthma (1.51, 1.04-2-21; p = 0.032).The associations were confirmed when using different sets of adjustment variables (Fig 2, models 1 and 2).

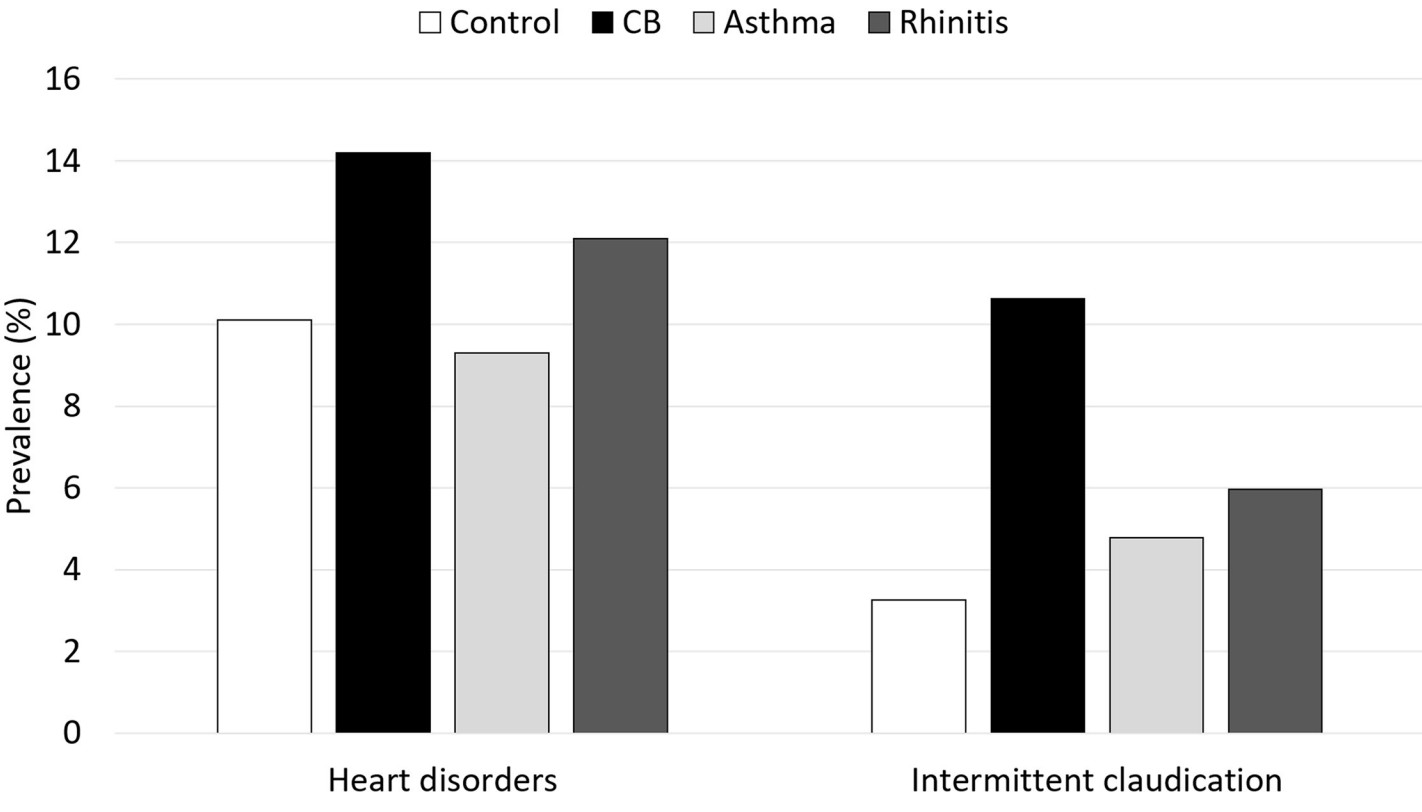

**Fig 1. Prevalence of heart disorders and intermittent claudication in subjects with CB, asthma, rhinitis, and controls.**

## Discussion

In the present analysis of data from the general population, we found that heart disorders and intermittent claudication, an indicator of symptomatic peripheral artery disease, were both strongly associated with CB and, to a minor extent, with rhinitis; intermittent claudication was also associated with asthma. Our data also indicate that cardiovascular risk factors are frequently associated with respiratory illnesses. Particularly, our findings support that unhealthy lifestyles (smoking, high alcohol consumption and sedentariness), hypertension and dyslipidaemia may predict a greater risk of CB.

Our findings should be interpreted keeping in mind that, in this multicase-control study, we used a hierarchical classification of diseases, so that cases of chronic bronchitis may also suffer from asthma and rhinitis, and cases of asthma could also be affected by rhinitis. Furthermore we underline that control subjects were accurately selected on the basis of the absence of CB, asthma and rhinitis.

CB was strongly and independently associated with heart disorders. Of note, as cases of COPD were excluded from the analysis, all subjects reporting CB presented a preserved lung function. Our results are in agreement with previous studies, demonstrating an increased risk of coronary disease and mortality among subjects with symptoms of CB [4,27,28]. In the above mentioned studies, lung function test was not performed, so that CB population could include subjects with COPD, which is known to be associated with cardiovascular diseases [2]. Lange et al. [5] and Guerra et al.[3] demonstrated an association between CB without bronchial obstruction, and all-cause death, indirectly supporting the results from the present analysis.

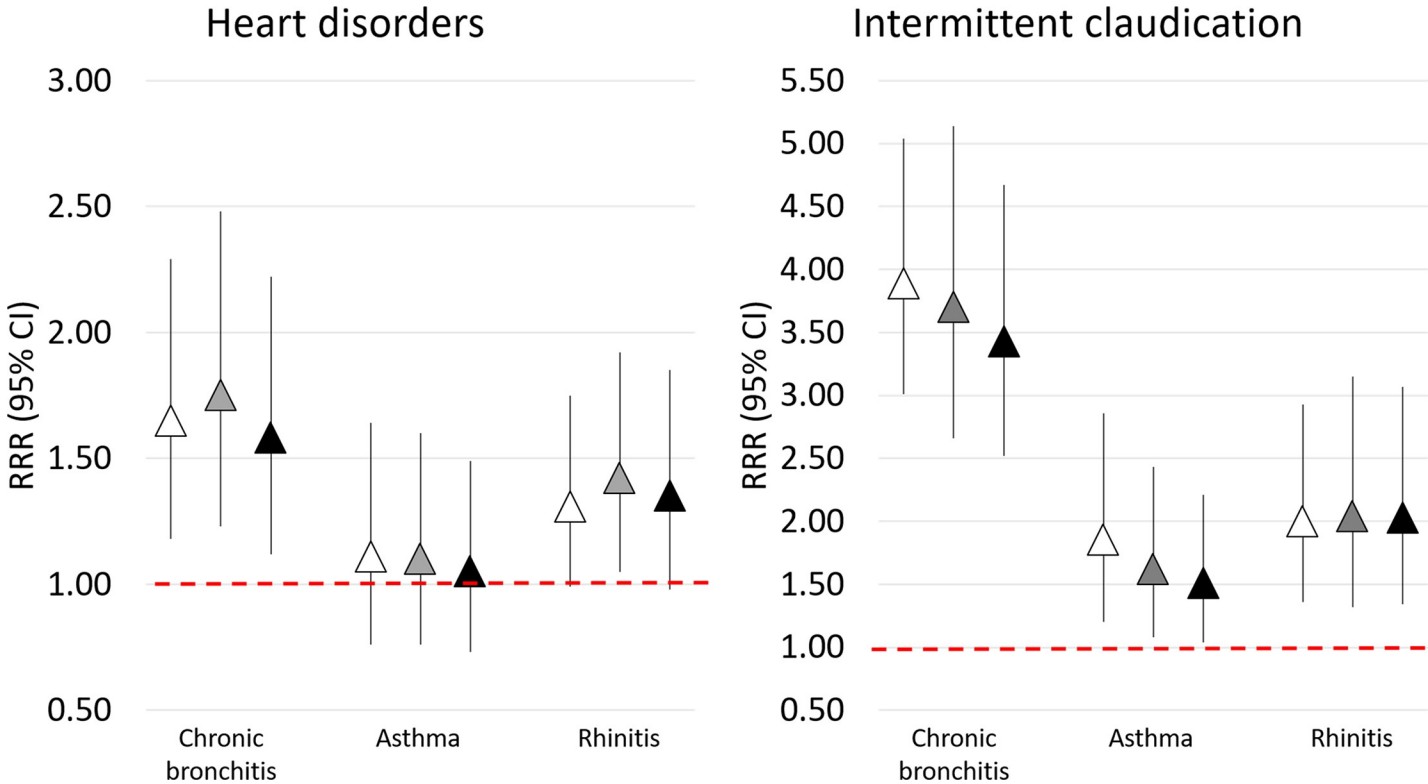

**Fig 2. Relative risk ratios (RRR) with 95% CIs for the associations of CB, asthma, and rhinitis with heart disorders and intermittent claudication.** Model 1 (white triangles): adjusted for age and gender; model 2 (grey triangles): adjusted for age, gender, school education, smoking status, alcohol consumption, BMI, physical activity; model 3 (black triangles): adjusted for variables in model 2 plus diabetes, hypertension, dyslipidaemia.

This study is the first to report an association between CB and intermittent claudication, which remained significant after controlling for all the most important cardiovascular risk factors. These results suggest that CB may be an independent risk factor for atherosclerosis.

The lack of a significant association between asthma and heart disorders is in agreement with Schanen and colleagues, who reported that asthma is not a risk factor for coronary heart disease [10]. Nonetheless, some reports suggested an association of asthma with carotid atherosclerosis, coronary heart disease and stroke [9–14]. One possible explanation for the contrasting results is the fact that, differently from previous studies [9–12], we analysed asthma separately from CB, a disorder we found strongly associated with cardiovascular risk factors and diseases. Furthermore, in our study asthma was precisely characterized, whereas it was self-reported in some previous investigations [13], a fact that could introduce a bias of misclassification with COPD [29].

Extending the finding of our previous analysis that was carried out using data from the centre of Verona alone [16], subjects with rhinitis and/or asthma had an increased risk of suffering from intermittent claudication.

Few studies investigated the relationship between cardiovascular diseases and rhinitis [17,30]. In disagreement with our results, Hirsch et al. did not find a significant association between chronic rhino-sinusitis and post-morbid cardiovascular conditions [31].

The strong association of smoking with CB is not surprising [32], but we also found an increased risk of asthma among current smokers, in line with findings from others [33,34].

Our results do not support previous findings suggesting a relationship between smoking and prevalence of chronic rhinitis [35].

We found an association between dyslipidaemia and CB. Subjects with CB are more likely to be heavy smokers and alcohol consumers, which may be responsible of an altered lipid metabolism [36,37]. However, the association between dyslipidaemia and CB persisted after controlling for these confounders.

Another interesting result is that subjects with rhinitis were less likely to be overweight or obese, which was also observed in other two recent surveys [38,39].

Differently from previous studies showing an association between obesity and the risk of asthma [40], we found no relationship between asthma and BMI.

In agreement with previous studies [41], elevate alcohol consumption was independently correlated with CB. Alcohol acts systemically with various mechanisms (alteration of immunity and promotion of systemic inflammation) [42], that may be involved in both the respiratory and cardiovascular damage.

Our finding of a relationship between rhinitis and hypertension is consistent with a previous study by Kony et al. [43]. Our data also indirectly support another case-control study that suggested an increased incidence of hypertension among subjects with rhino-sinusitis [31]. While the association between COPD and hypertension has widely been described [2], our study is the first to report a strong positive association between CB and this risk factor. Our data also show the increased risk of arterial hypertension among asthmatic subjects, in agreement with previous findings from large population based studies [9,44].

The nature of the association of cardiovascular disorders with rhinitis and CB remains speculative and several mechanisms, such as infection [45–47] and inflammation, may play a role. The association of atherosclerosis with respiratory diseases could also be caused by the inherent susceptibility of some subjects to specific inflammatory pathways. There is evidence that patients with rhino-sinusitis [48] and CB [3] have higher levels of C-reactive protein, a predictive marker of coronary heart disease [49].

Asthma and CB are both characterized by chronic inflammation in the lung, even though the nature of the inflammation differs between the two disorders [8,50]. The different types of inflammation probably result in distinct pathology, clinical manifestation [8], and could differently influence the development of cardiovascular comorbidities.

Finally, reversal causation could not be excluded, since our study design does not consent to assess the temporal relationship between respiratory disorders, cardiovascular diseases and risk factors. Subjects affected by cardiovascular disease and hypertension often use medication that might induce respiratory symptoms (such as cough) or disorders (such as rhinitis) [51]. However the association between rhinitis, CB, asthma and hypertension was not modified after adjusting for antihypertensive treatment (data not shown).

The strength of our analysis is based on the standardized protocol which allowed a precise definition of each respiratory disease. A limit is that cardiovascular events were self-reported. However, a previous study from the general population showed a good sensitivity and specificity for self-reported diagnosis of cardiovascular events [24].

We conclude that a better understanding of the relationship between respiratory and cardiovascular diseases could have important clinical implications. First of all, CB, which is often considered as a minor symptom, has to be viewed as a status possibly evolving not only to irreversible airway obstruction [3] but also to cardiovascular damage. Secondly, there are possible consequences for disease management, such as screening, prevention and early treatment of cardiovascular diseases and risk factors in patients with chronic bronchitis, even in absence of irreversible airflow obstruction. In turn, in patients with cardiovascular diseases caution should be adopted about the use of drugs that could negatively interfere with the respiratory system (e.g. angiotensin-converting enzyme inhibitors potentially inducing cough).

In our study, cases of CB may also suffer from asthma and rhinitis. Thus our data suggest the crucial weight of cough and phlegm in driving the association between respiratory and cardiovascular risk factors and diseases. As a matter of fact, the strength of the associations was lower among subjects with asthma or rhinitis who did not complain of CB. This attention to CB may have also a strong preventive consequence, since the disease is generally present even without a clinically significant airway derangement. A similar consideration is also of importance for rhinitis, taking into account its association with peripheral arterial disease.

Finally, although the design of our study does not allow definitive conclusions, we are tempted to speculate that some cardiovascular risk factors, such as sedentariness, hypertension and dyslipidaemia, might also be involved in the development of respiratory diseases.

## Supporting information

**S1 Dataset. Minimal anonymised data set to replicate the analyses.**
(CSV)

## Author Contributions

**Conceptualization:** Marcello Ferrari, Elia Piccinno, Alessandro Marcon, Giancarlo Pesce.

**Data curation:** Alessandro Marcon, Pierpaolo Marchetti, Giancarlo Pesce.

**Formal analysis:** Alessandro Marcon, Pierpaolo Marchetti, Giancarlo Pesce.

**Funding acquisition:** Marcello Ferrari, Alessandro Marcon, Lucia Cazzoletti, Pietro Pirina, Salvatore Battaglia, Leonardo Antonicelli, Giuseppe Verlato.

**Investigation:** Marcello Ferrari, Elia Piccinno, Alessandro Marcon, Giancarlo Pesce.

**Methodology:** Marcello Ferrari, Elia Piccinno, Alessandro Marcon, Giancarlo Pesce.

**Resources:** Marcello Ferrari, Alessandro Marcon, Lucia Cazzoletti, Pietro Pirina, Salvatore Battaglia, Amelia Grosso, Giulia Squillacioti, Leonardo Antonicelli, Giuseppe Verlato.

**Supervision:** Marcello Ferrari.

**Visualization:** Giancarlo Pesce.

**Writing – original draft:** Marcello Ferrari, Elia Piccinno, Alessandro Marcon, Giancarlo Pesce.

**Writing – review & editing:** Marcello Ferrari, Elia Piccinno, Alessandro Marcon, Pierpaolo Marchetti, Lucia Cazzoletti, Pietro Pirina, Salvatore Battaglia, Amelia Grosso, Giulia Squillacioti, Leonardo Antonicelli, Giuseppe Verlato, Giancarlo Pesce.

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
