## [Decision Letter · Decision Letter 0]

1 Oct 2019

PONE-D-19-23787

Chronic bronchitis without airflow obstruction, asthma and rhinitis are differently associated with cardiovascular risk factors and diseases

PLOS ONE

Dear Dr Alessandro Marcon,

Thank you for submitting your manuscript to PLOS ONE. After careful consideration, we feel that it has merit but does not fully meet PLOS ONE’s publication criteria as it currently stands. Therefore, we invite you to submit a revised version of the manuscript that addresses the points raised during the review process.

We would appreciate receiving your revised manuscript by December 1st 2019. To enhance the reproducibility of your results, we recommend that if applicable you deposit your laboratory protocols in protocols.io, where a protocol can be assigned its own identifier (DOI) such that it can be cited independently in the future. For instructions see: http://journals.plos.org/plosone/s/submission-guidelines#loc-laboratory-protocols

We look forward to receiving your revised manuscript.

Kind regards,

Davor Plavec

Academic Editor

PLOS ONE

Journal Requirements:

1. Thank you for inlcuding your ethics statement; "Ethical approval was obtained in each centre from the appropriate ethics committee, and written consent was obtained from each participant (see Acknowledgements)."

Additional Editor Comments (if provided):

Please submit the revised manuscript according to the suggestions of reviewer.

Reviewers' comments:

Reviewer's Responses to Questions

**Comments to the Author**

1. Is the manuscript technically sound, and do the data support the conclusions?

Reviewer #1: Partly

2. Has the statistical analysis been performed appropriately and rigorously? 

Reviewer #1: I Don't Know

3. Have the authors made all data underlying the findings in their manuscript fully available?

Reviewer #1: Yes

4. Is the manuscript presented in an intelligible fashion and written in standard English?

Reviewer #1: Yes

5. Review Comments to the Author

Reviewer #1: My major concern is a definition of the groups. For CB from the text I understand that their lung function was normal. That should be stated in the definition. Also since they were tested to allergies and metacholine was that an exclusion criteria? If yes that should be stated in the definition and an explanation of what caused CB since they were younger and 44% non-smokers? Asthma group shouldn’t be defined by 1. Criteria, only the second criteria is excluding non asthmatics from the group. Where are asthma and CB patients with rhinitis? Are they in asthma and CB groups (should be stated how much) or excluded?

6. PLOS authors have the option to publish the peer review history of their article (what does this mean?). If published, this will include your full peer review and any attached files.

Reviewer #1: No

---

## [Author Response · Author response to Decision Letter 0]

23 Oct 2019

PONE-D-19-23787

Chronic bronchitis without airflow obstruction, asthma and rhinitis are differently associated with cardiovascular risk factors and diseases

POINT-BY-POINT REPLY TO COMMENTS

Due date: December 1st 2019

Journal Requirements

We have revised the style of the manuscript to fulfil PLOS ONE requirements, including title page requirements, labelling of Figures, file naming, and reference formatting.

2) Thank you for including your ethics statement; "Ethical approval was obtained in each centre from the appropriate ethics committee, and written consent was obtained from each participant (see Acknowledgements)." Please amend your current ethics statement to include the full name of the ethics committee/institutional review board(s) that approved your specific study. Once you have amended this/these statement(s) in the Methods section of the manuscript, please add the same text to the “Ethics Statement” field of the submission form (via “Edit Submission”).

We have implemented this change in the manuscript (lines 106-112 of the tracked-changes version) and amended the ethics statement in the online submission form. 

3) We note that you have indicated that data from this study are available upon request. PLOS only allows data to be available upon request if there are legal or ethical restrictions on sharing data publicly. For information on unacceptable data access restrictions, please see http://journals.plos.org/plosone/s/data-availability#loc-unacceptable-data-access-restrictions. 

We have included a “minimal anonymised data set” as an additional online supplementary file (S1 Dataset), and updated our data availability statement accordingly.

We have checked the figure files using PACE as requested. 

Reviewers' comments

1) My major concern is a definition of the groups. For CB from the text I understand that their lung function was normal. That should be stated in the definition. 

Thanks for your comment. We have revised the definition of chronic bronchitis as suggested (line 132 in the tracked-changes version of the manuscript).

2) Also since they were tested to allergies and metacholine was that an exclusion criteria? If yes that should be stated in the definition and an explanation of what caused CB since they were younger and 44% non-smokers? 

As reported in the Methods section, we adopted a hierarchical classification of cases (lines 259-261). As a consequence, chronic bronchitis has cough and phlegm without bronchial obstruction as a hallmark, but in this group also subjects with asthma, rhinitis or atopy may be present. Positive skin prick tests or positive methacholine challenge were not exclusion criteria. 

In the group of subjects with chronic bronchitis, the symptoms may be related to causes other than smoking, including e.g. post-nasal drip and airway inflammation of asthma. Moreover, we cannot rule out that the symptoms can be linked to air pollution, which is a major issue in the Po Valley (Verona, Pavia and Turin centres). As highlighted in the manuscript, our findings suggest the critical clinical relevance of cough and phlegm even in the absence of smoking.

3) Asthma group shouldn’t be defined by 1. Criteria, only the second criteria is excluding non asthmatics from the group. 

“Definition 1” is “self-reported asthma, plus one among having had an asthma attack in the last 12 months or current use of medications for asthma”. This epidemiological disease definition has been widely used, and validated in studies demonstrating an excellent specificity, 99.7% (de Marco R et al. Eur Respir J. 1998;11: 599-605). 

4) Where are asthma and CB patients with rhinitis? Are they in asthma and CB groups (should be stated how much) or excluded?

Due to our hierarchical definitions, subjects with rhinitis could be included either among cases of chronic bronchitis, or among cases of asthma, or among cases of rhinitis alone. This is explained in lines 131-140 of the manuscript. 

In the revised manuscript, we have added the following sentence to the results section (lines 192-194): “As a result of our hierarchical definitions, cases of chronic bronchitis could also be affected by asthma (n=115, 45.3%) as well as rhinitis (n=205, 80.7%), and cases of asthma could also be affected by rhinitis (n=354, 84.7%)”.

We thank the reviewer for the time he/she dedicated in reviewing the manuscript.

---

## [Editor Report · Decision Letter 1]

28 Oct 2019

Chronic bronchitis without airflow obstruction, asthma and rhinitis are differently associated with cardiovascular risk factors and diseases

PONE-D-19-23787R1

Dear Dr. Alessandro Marcon,

We are pleased to inform you that your manuscript has been judged scientifically suitable for publication and will be formally accepted for publication once it complies with all outstanding technical requirements.

With kind regards,

Davor Plavec

Academic Editor

PLOS ONE

Additional Editor Comments (optional):

After suggested revision the manuscript is acceptable for publication in its current form.
---

## [Editor Report · Acceptance letter]

31 Oct 2019

PONE-D-19-23787R1 

Chronic bronchitis without airflow obstruction, asthma and rhinitis are differently associated with cardiovascular risk factors and diseases 

Dear Dr. Marcon:

I am pleased to inform you that your manuscript has been deemed suitable for publication in PLOS ONE. Congratulations! Your manuscript is now with our production department. 

With kind regards,

on behalf of

Dr. Davor Plavec 

Academic Editor

PLOS ONE